# Illuminating Hypomania in Early Adolescence: Associations Between Dark-Side and Bright-Side Hypomania, Insomnia, and Health-Related Quality of Life

**DOI:** 10.3390/jcm13226785

**Published:** 2024-11-11

**Authors:** Larina Eisenhut, Dena Sadeghi-Bahmani, Kenneth M. Dürsteler, Thorsten Mikoteit, Christian Fichter, Annette Beatrix Brühl, Zeno Stanga, Serge Brand

**Affiliations:** 1Center for Affective, Stress and Sleep Disorders (ZASS), Psychiatric University Hospital Basel, 4002 Basel, Switzerland; larina.eisenhut@unibas.ch (L.E.); annette.bruehl@upk.ch (A.B.B.); 2Department of Psychology, Stanford University, Stanford, CA 94305, USA; bahmanid@stanford.edu; 3Department of Epidemiology & Population Health, Stanford University, Stanford, CA 94305, USA; 4Psychiatric Clinics, Division of Substance Use Disorders, University of Basel, 4002 Basel, Switzerland; kenneth.duersteler@upk.ch; 5Center for Addictive Disorders, Department of Psychiatry, Psychotherapy and Psychosomatics, Psychiatric Hospital, University of Zurich, 8057 Zurich, Switzerland; 6Psychiatric Services Solothurn, University of Basel, 4503 Solothurn, Switzerland; thorsten.mikoteit@unibas.ch; 7Department of Psychology, Kalaidos Private University of Applied Sciences, 8050 Zurich, Switzerland; christian.fichter@kalaidos-fh.ch; 8Centre of Competence for Military and Disaster Medicine, Swiss Armed Forces, 3008 Bern, Switzerland; zeno-giovanni.stanga@vtg.admin.ch; 9Division of Diabetes, Endocrinology, Nutritional Medicine and Metabolism, University Hospital and University of Bern, 3010 Bern, Switzerland; 10Sleep Disorders Research Center, Kermanshah University of Medical Sciences, Kermanshah 6714869914, Iran; 11Substance Abuse Prevention Research Center, Health Institute, Kermanshah University of Medical Sciences, Kermanshah 6714869914, Iran; 12Department of Sport, Exercise and Health, Division of Sport Science and Psychosocial Health, University of Basel, 4052 Basel, Switzerland; 13School of Medicine, Tehran University of Medical Sciences, Tehran 1417466191, Iran; 14Center for Disaster Psychiatry and Disaster Psychology, Psychiatric University Hospital Basel, 4002 Basel, Switzerland

**Keywords:** hypomania, early to mid-adolescence, bright-side hypomania, dark-side hypomania, insomnia, health-related quality of life

## Abstract

**Background:** Adolescence is a period of significant psychological, physical, and social changes. During this time, adolescents face increasing responsibilities, such as making educational and career decisions, managing peer relationships, and becoming more independent from their families. These changes are often accompanied by mood fluctuations and altered sleep patterns. This study aimed to explore the relationships between bright- and dark-side hypomania, insomnia, and various dimensions of health-related quality of life (HRQOL), such as self-esteem, family and peer relationships, social acceptance, and autonomy. **Methods:** A total of 1475 participants in mid-adolescence (mean age: 13.4 years; range: 11–16 years; 48.8% males) completed a series of self-reported questionnaires covering sociodemographic information, hypomania, including dark and bright-side hypomania, insomnia, and HRQOL. **Results:** Compared to participants with no or dark-side hypomania, participants with bright-side hypomania reported better HRQOL. Bright-side hypomania was significantly associated with favorable relationships with parents and home, peer relationships, and the school environment and with less insomnia. In contrast, dark-side hypomania showed significant associations with lower scores for self-esteem, moods and emotional states, peer relationships, social acceptance, the school environment, and more insomnia. **Conclusions:** Among a larger sample of adolescents, bright- and dark-side hypomania were associated with a broad, though specific variety of aspects of HRQOL and insomnia. Given that standardized programs are available to improve insomnia and resilience as a proxy of psychological well-being, such interventions may have the potential to improve adolescents’ psychological well-being and sleep quality concomitantly.

## 1. Introduction

Adolescence is a period of significant development and changes in emotion, cognition, behavior, and social behavior, which are concomitantly associated with psychophysiological changes including neuronal, and neuroendocrine alterations [1,2]. During adolescence, various developmental and contextual transitions converge, meaning that adolescents are simultaneously and in quick succession confronted with numerous changes in both their personal psychosocial and physiological development and social environment [3]. Not surprisingly, the exposure to a variety of challenges during adolescence such as academic pressure, relationships with peers and parents, and identity formation has the potential to be highly associated with or to influence an adolescent’s self-perception and psychological well-being [4,5].

### 1.1. Adolescence and Mood Swings

Moreover, adolescence is a critical period for the onset of mood disorders. Approximately half of all behavioral and emotional disorders emerge during late childhood and early adolescence [6,7,8], and such behavioral and emotional alterations increase the risk of developing mental issues later in life [9,10,11,12]. Specifically, according to Lancet [13], mood and anxiety disorders accounted for 45% of the global burden of disease in adolescents, highlighting the substantial impact these conditions have within this age group.

In parallel with psychosocial changes, adolescence is also characterized by remarkable changes in neurocognitive functioning, particularly in domains where alterations associated with mood disorders have been observed [14,15,16]. These neurocognitive changes include emotion processing, impulse control, and decision-making [17]. During adolescence, the brain regions responsible for self-regulation, planning, and impulse control, primarily located in the prefrontal cortex, develop more slowly than the reward-sensitive areas of the limbic system. This developmental disparity can lead to a phase where the urge for immediate rewards outweighs self-control. Consequently, abnormal development in these neurocognitive regions may lead to the presence of symptoms of mood disorders such as symptoms of bipolar disorders and major depressive disorder [17,18]. This implies that aberrant development of neurocognition may be tightly associated with bipolar and unipolar mood disorders [19,20].

### 1.2. Adolescence and Bipolar Disorders

In recent years, there has been growing attention on the field of pediatric and adolescent bipolar disorder (BD) due to its high morbidity rates [21]. Bipolar disorder is a complex psychiatric disorder, characterized by recurring episodes of mania or hypomania and depression, with alternating periods of euthymia [22,23,24]. Approximately 27.7% of individuals experienced the first onset of bipolar disorder (BD) during childhood, before the age of 13 years, while an additional 37.6% manifested symptoms of BD during adolescence, between the ages of 13 and 18 [25]. Despite this, the average age of onset for BD was typically between 18 and 22 years [14], indicating a wide range of onset ages among individuals with the disorder.

The estimated global lifetime prevalence for bipolar I disorder ranges between 0.6 and 1%, and between 0.4 and 1.1% for bipolar II disorders [26,27]. More recently, for youth, the term bipolar spectrum disorders was introduced, an umbrella term that covers bipolar I disorder, bipolar II disorder, cyclothymic disorder (CycD), bipolar disorder not otherwise specified (NOS), and other specified bipolar and related disorders (OS-BRD), leading to higher prevalence rates of 2.06% [28]. Clearly, broader definitions (i.e., including NOS or CycD) were associated with higher prevalence rates than narrow definitions (i.e., BD-I and -II only). When considering the narrow definition, prevalence rates range from 0.29% to 0.80% [29].

In clinical settings, both BD type I (BD-I) and BD type II (BD-II) are frequently misdiagnosed as different psychiatric conditions [30,31]. BD-I is defined by episodes of mania, whereas BD-II involves only hypomanic episodes [32] in combination with the occurrence of depressive disorders. However, BD-II typically exhibits more frequent and prolonged depressive episodes and a greater chronicity of illness compared to BD-I [33,34]. Given the seriousness and widespread occurrence of bipolar disorder (BD), there is an increasing focus on investigating hypomanic symptoms in children and adolescents. Enhancing our comprehension of how hypomania manifests during adolescence can consequently enhance the precision and timeliness of diagnosing BD [35].

Further, according to DSM-5-TR [32] a hypomanic episode involves a distinct period of unusually and persistently elevated, expansive, or irritable mood and unusually or persistently increased activity or energy, lasting at least four consecutive days. The core features, as defined by DSM-5 TR, include: 1. Inflated self-esteem or grandiosity; 2. Decreased need for sleep; 3. More talkative than usual; 4. Flight of ideas or subjective experience that thoughts are racing; 5. Distractibility; 6. Increase in goal-directed activity; 7. Excessive involvement in activities that have a high potential for painful consequences. Mood disturbances and changes in functioning are observable by others, but hypomanic episodes, by DSM-5-TR definition, are not severe enough to cause marked impairment in social or occupational functioning or to necessitate hospitalization.

In a similar vein, assessing dimensions of hypomania, including its sub-categories of dark-side and bright-side hypomania, is considered a non-clinical approach to mood swings as a proxy of symptoms of bipolar alterations [36].

Hantouche et al. [36] initially classified hypomania into what they termed the “bright” and “dark” sides. The bright side encompasses socially positive and advantageous attributes such as creativity increased energy, enhanced self-esteem, and talkativeness often referred to as active-elated. Conversely, the dark side is characterized by socially negative aspects such as impulsivity, risk-taking behavior, concentration difficulties, and aggression, and is also referred to as the irritability/risk-taking dimension [37]. This categorization has provided a framework for understanding the multifaceted nature of hypomania. According to Holtmann et al. [38], in a sample of 294 non-clinical adolescents (average age 17.3 years, 47.6% females), about 11.2% exhibited hypomanic symptoms as assessed by the Hypomania Checklist 32 (HCL-32: [39] ). Higher scores for bright-side hypomania were associated with lower scores for peer issues, while higher scores for dark-side hypomania were associated with higher scores for behavioral problems, hyperactivity-attention issues, and peer difficulties [38]. In short, both positive (“bright”) and negative (“dark”) aspects of hypomania were associated with unique and district interpersonal and behavioral outcomes. In a study involving 862 young adults (M = 24.67; 74.13% females), participants scoring high on the “dark side” hypomania reported elevated levels of depressive symptoms, sleep disturbances, somatic complaints, perceived stress, negative coping strategies, and lower self-efficacy compared to non-hypomanic participants and compared to those in the “bright side” group. By contrast, individuals scoring high on the “bright side” of hypomania showed reduced stress levels, more positive self-instructions, and higher levels of exploration, self-efficacy, and physical activity [37]. Further, among young adults in love, bright-side hypomania was associated with reduced scores for depression and anxiety, and higher scores for improved sleep quality [40]. See also more recent research on the factor structure of the HCL-32, including the distinction between an active-elated and risk-taking/irritable-erratic factor model [35,41].

### 1.3. Sleep Patterns During Adolescence

There is extant evidence that adolescence not only marks a pivotal phase characterized by an increased prevalence of dimensions of psychopathology but also notable changes in sleep behavior. The somatic, neuronal, behavioral, and psychological characteristics of adolescence are associated with maturational changes in the sleep-wake cycle, sleep timing, sleep duration, and architecture of sleep [42].

Adolescents require approximately 8–10 h of sleep per night for optimal functioning [43,44,45]. Despite this demand, many adolescents fail to achieve sufficient sleep due to a combination of biological changes and external pressures, such as homework, extracurricular activities, and early school start times [46]. Earlier research indicated that insufficient sleep was associated with various adverse outcomes, including poor academic performance [47,48], increased risk of mental health issues [49,50], and behavioral problems [51,52].

Sleep deprivation was also linked to emotion regulation issues, leading to increased irritability, mood swings, and susceptibility to stress. Adolescents with chronic sleep problems were at a higher risk for developing anxiety and depression [53]. Additionally, research indicated that sleep disturbances were prevalent in individuals with mood disorders, including BD type I and II, and that such sleep disturbances were associated with the recurrence of mood episodes [54,55,56]. Brand et al. [37] demonstrated that young adults (M = 24.67, 74.13% females) currently in a phase of dark-side hypomania reported more sleep problems than those reporting bright-side hypomania or no hypomania at all. The bright-side hypomania and no hypomania groups showed no significant differences in sleep problems.

### 1.4. The Present Study

Hypomanic episodes, which often include a decreased need for sleep according to DSM-5-TR [32], can be associated with insomnia, which in turn may exacerbate symptoms and complicate the diagnosis and management of BD. Given that insomnia is prevalent among individuals with mood disorders, including BD, the relationship between hypomania and insomnia could be particularly relevant for adolescents. Surprisingly, there exists a notable gap in the research regarding the relationship between sleep, the bright and dark sides of hypomania, and dimensions of the health-related quality of life (HRQOL) in non-clinical adolescents. In this study, HRQOL was understood as a multidimensional construct, reflecting the physical, emotional, social, and self-perception aspects of well-being in children and adolescents. It captures how these factors collectively influence the overall quality of life. More specifically, HRQOL was defined to include physical and psychological health, emotional well-being, self-perception, autonomy, family relationships, financial circumstances, social support, the school environment, and social acceptance. The present study aimed to fill this research gap by formulating four hypotheses to investigate these associations. The following four hypotheses were formulated.

First, we expected significant group differences in various aspects of HRQOL among participants with high bright-side hypomania, high dark-side hypomania, and no hypomania. Specifically, we hypothesized that participants with high bright-side hypomania scores have higher HRQOL scores, whereas those with high dark-side hypomania scores have lower HRQOL scores. For participants with no hypomania, we expected higher HRQOL scores than the dark-side hypomania group, but lower scores than the bright-side group. This is supported by Brand et al. [37] who found that bright-side hypomania was associated with increased physical activity, lower stress levels, and better psychological functioning, while dark-side hypomania was linked to depressive symptoms and poor emotional well-being.

For the second hypothesis, we expected a significant association between bright- and dark-side hypomania and various aspects of HRQOL. Specifically, higher scores for bright-side hypomania were hypothesized to be associated with higher HRQOL scores, while higher scores for dark-side hypomania were expected to be associated with lower HRQOL scores. This is based on the idea that the positive attributes of bright-side hypomania, such as increased energy and elevated mood, may enhance daily functioning and social well-being, whereas the negative attributes of dark-side hypomania may impair physical and psychological well-being [37]. Gamma et al. [57] and Jahangard et al. [58] provide evidence supporting this, showing that bright-side hypomania is related to better social and physical activity levels and improved well-being, while dark-side hypomania is correlated with depression and stress.

For the third hypothesis, we anticipated significant group differences in insomnia among participants with high bright-side hypomania, high dark-side hypomania, and no hypomania. Specifically, we hypothesized that participants with high bright-side hypomania scores have lower insomnia scores, whereas those with high dark-side hypomania scores have higher insomnia scores. For participants with no hypomania, we expected lower insomnia scores than the dark-side hypomania group, but higher scores than the bright-side group.

Concerning our fourth hypothesis, following previous findings [37], we predicted a significant association between bright- and dark-side hypomania and insomnia. In detail, we expected no significant association between bright-side hypomania and insomnia, while higher scores for dark-side hypomania are expected to be associated with higher insomnia scores. This is in line with previous findings, suggesting that the dark side of hypomania is associated with sleep disturbances [37].

## 2. Methods

### 2.1. Participants

A total of 1475 adolescents (M = 13.4 years; range: 11–16 years; 48.8% males) participated in the study. They were recruited from five middle schools in the Cantons Basel, Basel-Land, and Aargau, three districts in the northwestern part of German-speaking Switzerland. After the approval of the ethical committee, deans of ten middle schools were contacted and asked whether we were allowed to run the study in their schools. Of the ten deans approached, five declined (main reasons: study did not fit with their syllabus and exam timetables), and five approved. Thereafter, we organized events to thoroughly inform adolescent students and their parents/legal guardians about the aims of the study, the confidential and secured data handling, including the full data protection in relation to the school management and teachers, and the assurance that participation or non-participation had no advantages or disadvantages for a student’s academic performance.

Next, students interested in participating in the study received the booklet of questionnaires, which was filled out in about 30 to 35 min. Once a questionnaire was completed, the form was handed back in a sealed envelope. During data collection in the classrooms, professional staff members provided examples of how to complete the items in the questionnaire booklet. The questionnaires covered sociodemographic information, insomnia, and psychosocial well-being and ill-being such as physical well-being, psychological well-being, moods and emotions, self-perception, autonomy, parent relations and home, financial resources, peers and social support, school environment, and bullying. Of the 1502 students approached, 1475 (98.02%) agreed to participate in the study. Inclusion criteria were: 1. Willing and able to comply with the study requirements. 2. The student and their legal guardian signed the written informed consent. Exclusion criteria: 1. Resign from study participation. 2. Blank forms. The study was performed in accordance with the Declaration of Helsinki. The Ethics Committee of Basel (EKNZ; Ethikkommission Nordwest- und Zentralschweiz; Basel, Switzerland; code: 72/10 2010; date of approval: 1 November 2010) approved the study. The study itself was performed during the spring semester of 2013.

### 2.2. Measures

#### 2.2.1. Hypomania

To assess the hypomanic state, participants completed the Hypomania Checklist-32 (HCL-32; [39] a self-assessment tool for hypomanic symptoms. The hypomanic state is assessed by “yes/no” responses to 32 statements concerning behavior (e.g., “I spend more money/too much money”), mood (e.g., “My mood is significantly better”), and thoughts (e.g., “I think faster”) within the last four weeks. In prior research, the questionnaire has demonstrated good psychometric properties and has repeatedly been shown accurately to assess hypomanic states [59]. The 32 items can be divided into two-factor analytically derived subscales, which differentiate between “active/elated” (“bright side”) hypomania and “risk-taking/irritable” (“dark side”) hypomania [37,60,61]. The validity of the two subscales has proved also to apply to non-clinical samples of both adults [62] and adolescents [38].

For this analysis, a shortened version of the HCL was used. Eight items were selected based on their high factor loadings for bright- and dark-side hypomania, and their favorable item difficulty, as documented elsewhere [37] (Cronbach’s alpha: 0.83).

To more accurately explore the differences between the bright and dark sides of hypomania, participants were classified into three groups based on the 75th percentile of their HCL scores. Participants with scores on the bright-side hypomania subscale at or above the 75th percentile were categorized into the “BRIHYP” group (High Bright-Side Hypomania). Similarly, those with scores on the dark-side hypomania subscale at or above the 75th percentile were classified into the “DAHYP” group (High Dark-Side Hypomania). Participants who did not meet the 75th percentile criteria on either dimension were placed in the “NOHYP” group (No Hypomania).

For hypotheses one and two, the three groups (BRIHYP, DAHYP, and NOHYP) were used for the analysis, whereas for hypotheses three and four, the individual scores for bright- and dark-side hypomania were utilized.

#### 2.2.2. Health-Related Quality of Life (HRQOL)

Participants completed the KID-SCREEN-52 questionnaire [63], to assess various aspects of their HRQOL. The questionnaire consists of 52 items that assess ten different scales related to children’s and adolescents’ social, physical, and psychological functioning. These scales cover areas such as physical and psychological well-being, emotions, self-perception, autonomy, family relationships, finances, social support, school environment, and social acceptance. The ten scales of the Kidscreen-52 can be further aggregated into five main domains: personality traits, emotional and mental health, social relationships, environmental and contextual factors, and physical health and well-being.

Responses are given on a 5-point Likert scale ranging from 1 (= “not at all”) to 5 (= “extremely/always”), with higher sum scores reflecting a more accentuated dimension. Further, a global health-related quality of life index is also calculated, with higher mean scores indicating better functioning (Cronbach’s alpha for the overall index = 0.92).

#### 2.2.3. Insomnia

The Insomnia Severity Index (ISI) [64] is a 7-item screening measure for insomnia and an outcome measure for use in treatment research. The items, answered on a 5-point rating scale (0 = not at all; 4 = very much), refer to the DSM-5 TR [32] criteria for insomnia by measuring difficulty in falling asleep, difficulties remaining asleep, early morning awakenings, increased daytime sleepiness, impaired daytime sleepiness, impaired daytime performance, low satisfaction with sleep, and worrying about sleep. The higher the overall score, the more the respondent is assumed to suffer from insomnia (Cronbach’s alpha = 0.89).

### 2.3. Statistical Analysis

First, group differences in HRQOL and insomnia were assessed. To evaluate the group differences in HRQOL, a multivariate analysis of variance (MANOVA) was performed, with the 10 HRQOL scales as the dependent variable and the groups (BRIHYP, DAHYP, NOHYP) as the independent factor. To assess the group differences for insomnia, an analysis of variance (ANOVA) was conducted. Post hoc tests using the Bonferroni–Holm correction were applied for both analyses.

Second, a series of Pearson correlations were performed between hypomania, insomnia, and HRQOL. Subsequently, two multiple regression analyses were performed to identify which aspects of HRQOL, including insomnia, were more strongly associated with dark- and bright-side hypomania.

Preliminary conditions to perform multiple regression analyses were met [65,66]: N = 1475 > 100; predictors explained the dependent variables (R = 0.389, R^2^ = 0.151 for dark-side hypomania in HRQOL; R = 0.294, R^2^ = 0.086 for dark-side hypomania in insomnia; R = 0.268, R^2^ = 0.072 for bright-side hypomania in HRQOL; R = 0.025, R^2^ = 0.001 for bright-side hypomania in insomnia); the number of predictors: 10 (for HRQOL) and 1 (for insomnia); 10 × 10 = 100 < N (1475) for HRQOL; 1 × 1 = 1 < N (1475) for insomnia; and the Durbin–Watson coefficient was 1.989 for dark-side hypomania in HRQOL, 1.950 for dark-side hypomania in insomnia, 1.964 for bright-side hypomania in HRQOL, and 1.935 for bright-side hypomania in insomnia, indicating that the residuals of the predictors were independent. Furthermore, the variance inflation factors (VIF) were between 1.201 and 3.973; while there are no strict cut-off points to report the risk of multicollinearity, VIF < 1 and VIF > 10 indicate multicollinearity (Hair 2014 [66]). The nominal significance level was set at alpha < 0.05. Statistics were performed using SPSS^®^ 29.0 (IBM Corporation, Armonk, NY, USA) for Apple Mac^®^ (Apple Inc., Cupertino, CA, USA).

## 3. Results

### 3.1. Differences in Health-Related Quality of Life Between High Bright-Side Hypomania, High Dark-Side Hypomania, and No Hypomania Groups

Table 1 reports an overview of the descriptive and inferential statistics of the three groups for HRQOL and insomnia. The BRIHYP group reported better HRQOL compared to the other two groups across all scales. The BRIHYP group exhibited higher scores for self-esteem, psychological well-being, enhanced psychological well-being, and more positive emotional states. Additionally, the BRIHYP group reported higher scores for more positive social relationships with both family and peers. The school environment and financial prospects were perceived more favorably, and overall physical health was better.

In contrast, individuals in the DAHYP group showed poorer outcomes across all HRQOL scales than the BRIHYP and NOHYP groups. Further, participants in the NOHYP group showed intermediate results between the BRIHYP and DAHYP groups. While the NOHYP group generally had better outcomes than the DAHYP group, their scores were lower than those of the BRIHYP group.

The MANOVAs confirmed these results, showing significant differences between the three groups regarding HRQOL. The post hoc tests after Bonferroni–Holm corrections also revealed significant differences across all HRQOL scales between the BRIHYP and DAHYP group, as well as between the NOHYP and DAHYP group. When comparing the BRIHYP and NOHYP groups, significant differences were observed regarding autonomy, peer relationships, social acceptance, school environment, and physical well-being.

### 3.2. Differences in Insomnia Between High Bright-Side Hypomania, High Dark-Side Hypomania, and No Hypomania Groups

The DAHYP group reported overall higher levels of insomnia. The ANOVA confirmed these findings, indicating a significant difference in insomnia scores between the three groups (Table 1).

### 3.3. Associations Between Dark- and Bright-Side Hypomania and Health-Related Quality of Life

Table 2 reports the descriptive statistics and the correlation coefficients (Pearson’s correlations) between the bright and dark sides of hypomania and HRQOL.

Higher scores for bright-side hypomania were associated with higher scores for self-esteem, autonomy, psychological well-being, moods, and emotional states, peer relationships, school environment, financial prospects, and physical well-being. However, higher scores for bright-side hypomania were not associated with relationships with parents and home and social acceptance. In contrast, higher scores for dark-side hypomania were associated with lower scores for self-esteem, autonomy, psychological well-being, moods and emotions, relationships with parents and home, peer relationships, social acceptance, school environment, financial prospects, and physical well-being. Thus, higher scores for dark-side hypomania were associated with lower scores for all dimensions of the health-related quality of life.

### 3.4. Associations Between Dark- and Bright-Side Hypomania and Insomnia

As shown in Table 2, scores for bright-side hypomania were statistically uncorrelated with insomnia. In contrast, higher scores for dark-side hypomania were significantly associated with higher insomnia scores.

### 3.5. Multiple Regression Analyses: Relationships Between Hypomania, Health-Related Quality of Life, and Insomnia

#### 3.5.1. Hypomania and Health-Related Quality of Life

Table 3 reports the overview of the multiple regression models between the bright and dark sides of hypomania and HRQOL and insomnia. Table 4 presents the multiple regression with the bright side of hypomania and the 10 different HRQOL scales. Table 5 shows the multiple regression with the dark side of hypomania and the 10 different HRQOL scales.

The multiple regression models (Table 3) revealed that HRQOL was significantly associated with bright-side hypomania (F(10, 1465) = 11.396, *p* < 0.001, n = 1475), explaining 6.6% of its variance (Table 3). Higher scores for bright-side hypomania were associated with higher scores in relationships with family and home, peer relationships, and school environment (Table 4).

Similarly, the multiple regression models also demonstrated a significant association between HRQOL and dark-side hypomania (F(10, 1465) = 26.103, *p* < 0.001, n = 1475), explaining 15.1% of its variance, indicating a moderate effect size according to Cohen. In detail, higher scores for dark-side hypomania were significantly associated with lower scores in self-esteem, moods and emotional states, social acceptance, and school environment, while showing a positive association with peer relationships (Table 5).

#### 3.5.2. Hypomania and Insomnia

The multiple regression analysis indicated that insomnia did not have a significant association with bright-side hypomania (F(1, 1474) = 1.473, *p* = 0.339). Conversely, insomnia was significantly associated with dark-side hypomania (F(1, 1474) = 136.451, *p* < 0.001), explaining 8.6% of its variance (Table 3). Higher scores for dark-side hypomania were associated with higher scores in insomnia.

## 4. Discussion

The aims of the present study were to examine the relationship between hypomania, insomnia, and HRQOL in a non-clinical sample of adolescents. The key findings were that both bright and dark sides of hypomania were associated with different aspects of HRQOL. The bright side of hypomania was associated with better social relationships and a higher overall quality of life, while the dark side, often accompanied by insomnia, was associated with lower self-esteem, poorer emotional states, poorer social acceptance, and lower overall quality of life.

Four hypotheses were formulated, and each of these is considered now in turn.

First, consistent with our first hypothesis, we found significant group differences in HRQOL. Adolescents with high bright-side hypomania showed significantly higher scores in autonomy, peer relationships, social acceptance, school environment, and physical well-being compared to participants with high dark-side hypomania or no hypomania. In contrast, the dark-side group exhibited significantly lower HRQOL across all dimensions compared to the other groups.

For the second hypothesis, we anticipated that bright-side hypomania is related to better HRQOL, while dark-side hypomania is associated with poorer HRQOL. The results, including both Pearson’s correlations and multiple regression analyses, supported this hypothesis. Regarding bright-side hypomania, Pearson correlations revealed significant positive associations between several aspects of HRQOL, including self-esteem, autonomy, psychological well-being, moods and emotions, peer relationships, school environment, and physical well-being. These findings suggest a broad positive association between bright-side hypomania and overall HRQOL. However, the multiple regression analysis provided a more nuanced view, identifying significant relationships between bright-side hypomania and relationships with parents and home, peer relationships, and school environment. This indicates that the strongest effects of bright-side hypomania appear to be in the social and environmental domains.

Concerning dark-side hypomania, Pearson correlations revealed significant negative associations with all aspects of HRQOL. These findings suggest a widespread negative association of dark-side hypomania with overall HRQOL. However, the multiple regression analysis identified key significant relationships between dark-side hypomania and self-esteem, moods and emotional states, peer relationships, social acceptance, and school environment.

These findings align with previous research [37] and imply that the positive characteristics of bright-side hypomania, such as heightened energy and greater social involvement, could lead to improved physical health, stronger social connections, and more favorable personality traits. The observed positive relationship between bright-side hypomania and HRQOL suggests that individuals with this form of hypomania may experience enhanced social interactions, boosted self-esteem, and a more optimistic view of their surroundings. This indicates that bright-side hypomania might involve adaptive strategies that promote resilience and contribute to overall well-being.

In contrast, the negative traits of dark-side hypomania, such as emotional instability, impulsiveness, and irritability, are linked to a poorer quality of life. These effects stem from increased risk-taking, mood swings, and difficulty controlling emotions, which disrupt various aspects of life. Emotional instability can lead to unpredictable moods and heightened stress, while impulsiveness may strain social interactions, leading to conflicts and challenges in maintaining relationships. Brand et al. [37] also found a strong association between dark-side hypomania and depressive symptoms in young adults. Considering the established connection between depression and reduced quality of life [67,68,69,70], our findings support these earlier studies. Moreover, dark-side hypomania can significantly affect social and environmental factors. Challenges in sustaining relationships with family and peers, along with academic difficulties, suggest that individuals with higher levels of dark-side hypomania may find it hard to maintain stable support systems. This could be due to irritability, aggression, or social withdrawal, which often accompany this form of hypomania. Additionally, problems in school, such as decreased focus or motivation, and declining academic performance may increase stress, exacerbating the emotional strain and further lowering the quality of life.

Both the bright and dark sides of hypomania emphasize the importance of social relationships and environmental factors. Positive interactions and a nurturing environment seem to enhance HRQOL for those experiencing bright-side hypomania, while the dark side tends to strain social connections and negatively impact the school environment, contributing to a lower HRQOL.

Interestingly, while the Pearson correlation indicated a negative relationship between dark-side hypomania and peer relationships, the multiple regression analysis showed a positive association. This seemingly contradictory result regarding the association between dark-side hypomania and peer relationships may be better understood by drawing parallels between dark-side hypomania and the traits of the “Dark Triad”. The Dark Triad is a combination of narcissism, Machiavellianism, and psychopathy. More specifically, the Dark Triad refers to a group of three distinct but related personality traits: narcissism [71] characterized by grandiosity and a need for admiration; Machiavellianism [72], marked by manipulation and exploitation of others; and psychopathy [73], which involves impulsivity, a lack of empathy, and antisocial behavior. Research has shown that individuals with high levels of Dark Triad traits often exhibit socially dominant, manipulative, and risk-taking behaviors, which can paradoxically lead to greater social success in certain contexts [74,75,76]. Despite emotional instability and impulsiveness, people with these traits may be more adept at navigating social environments and exerting influence over others. Though highly speculative, it is possible that individuals with dark-side hypomania might display similar behaviors, such as emotional volatility and risk-taking, which could lead to short-term social success, explaining the positive association with peer relationships. These behaviors may enhance their ability to manipulate social situations to their advantage, even if they also experience emotional and psychological difficulties in the long term.

For the third hypothesis, we anticipated differences in insomnia between the three groups. The results supported this hypothesis, revealing that adolescents with high scores for dark-side hypomania experienced significantly more insomnia compared to those with high bright-side hypomania scores or no hypomania.

In line with our fourth hypothesis, we expected no significant relationship between bright-side hypomania and insomnia, but a strong association between dark-side hypomania and insomnia. Both the Pearson correlations and multiple regression analyses confirmed this hypothesis. There was no association between bright-side hypomania and insomnia, while dark-side hypomania was significantly associated with increased insomnia.

These findings are consistent with previous research [37,77] and suggest that emotional instability, often seen in dark-side hypomania, might play a role in the development of sleep disturbances. Sleep issues may aggravate emotional dysregulation, worsening the challenges associated with dark-side hypomania. However, it is important to acknowledge that a reduced need for sleep is a diagnostic criterion for hypomania according to DSM-5 TR [32], and this aspect was not directly assessed in our study.

The lack of a significant relationship between bright-side hypomania and insomnia might suggest that individuals with bright-side hypomania are less susceptible to sleep disturbances, potentially due to higher resilience and mental toughness [58,78]. These qualities could enable individuals with bright-side hypomania to maintain better sleep, even with elevated energy levels or other typical hypomanic traits. Alternatively, this finding could be due to the set-up of the Insomnia Severity Index (ISI), which focuses solely on problematic sleep, without accounting for positive sleep quality or the complete absence of sleep issues. Consequently, variations in sleep behavior among individuals without sleep disturbances may not be captured, which could explain the lack of correlation.

### Limitations

While providing valuable insights, the study’s cross-sectional design presents challenges in establishing causal relationships. Moreover, engaging in longitudinal research to track these dynamics over time would substantially enhance our understanding. A longitudinal approach would help clarify whether more pronounced hypomanic traits observed in adolescents are relatively stable personality characteristics, or if they fluctuate as part of dynamic states. This distinction is particularly important, as these hypomanic states might serve as early signs or prodromal symptoms of bipolar disorder. Given the episodic nature of bipolar disorder, further investigation is necessary to understand how bright and dark hypomanic traits may develop over time and influence the progression toward clinical symptoms. It is also important to note that hypomanic states in adolescents can be triggered by intense emotional experiences, such as being “madly in love” [61], highlighting that not all hypomanic behavior necessarily indicates signs of psychopathology. Such contextual factors should be considered when interpreting hypomanic traits in this age group.

Another potential concern lies in the reliance on self-reported data, particularly for hypomania symptoms and sleep patterns. While we mentioned the use of self-reports in the study, it is crucial to acknowledge the limitations this method introduces, including response bias and potential over- or under-reporting. This calls for careful interpretation of the study’s findings. Furthermore, using a single measure, the Insomnia Severity Index, to assess sleep disturbances may oversimplify the complex landscape of sleep disturbances. Multiple measures, including actigraphy or polysomnography, could offer a more comprehensive and objective understanding of sleep patterns.

Next, the specificity of the sampled regions and schools could limit the generalizability of our results. Additionally, the age range of 11 to 16 years captures a spectrum of developmental stages, suggesting the need for further stratification. This would allow for a more nuanced exploration of how hypomania manifests in different adolescent age groups. Moreover, while the hypotheses are rooted in existing literature, the generalization of our findings to all adolescents may oversimplify the intricate nature of hypomania in this age cohort.

Additionally, the HCL-32 scale, used to measure hypomanic symptoms, has some limitations in non-clinical populations. Meyer et al. [62] noted that the standard cutoff of 14 might not be suitable for non-clinical samples, potentially leading to an overestimation of hypomania prevalence. Nonetheless, the HCL-32 has been successfully applied in non-clinical adolescent samples, as shown in previous studies [37,61].

Focusing exclusively on non-clinical adolescents also restricts the applicability of our results to clinical populations, such as those diagnosed with BD, who may exhibit hypomanic traits differently. Including clinical samples would provide a more comprehensive view of hypomania across the spectrum. Finally, relying solely on subjective self-reports to assess HRQOL highlights the need for objective measures or multi-informant assessments to strengthen the reliability of our findings.

## 5. Conclusions

Our findings illuminate the complex relationships between bright- and dark-side hypomania, HRQOL, and insomnia in non-clinical adolescents. Both the bright and dark sides of hypomania were associated with various aspects of HRQOL. Bright-side hypomania was associated with stronger social relationships, higher self-esteem, and a more positive school environment. In contrast, dark-side hypomania was associated with lower self-esteem, more negative emotional states, poorer peer relationships, lower social acceptance, and a more negative perception of the school environment, along with higher levels of insomnia.

These findings underscore how the different facets of hypomania can uniquely affect adolescent HRQOL. While the bright side may help promote resilience and better overall functioning, the dark side seems to negatively affect important social and emotional aspects.

Future research should approach longitudinal designs to better understand the causal dynamics between hypomania, HRQOL, and insomnia. Additionally, examining protective factors such as social support and coping strategies could offer insights into reducing the negative effects of hypomania, particularly the dark side, on HRQOL. This research could ultimately guide the development of targeted interventions and prevention programs aimed at improving mental health and quality of life for adolescents with hypomanic symptoms.

## Figures and Tables

**Table 1 jcm-13-06785-t001:** Overview of the descriptive and inferential statistics of the three groups BRIHYP (high bright-side hypomania), DAHYP (high dark-side hypomania), NOHYP (no high hypomania scores).

		Group								
		BRIHYP	DAHYP	NOHYP	Total			Post Hoc Tests (Bonferroni) *p*-Value		
Health-Related Quality of Life		496 M (SD)	189 M (SD)	790 M (SD)	1475 M (SD)	F	*p*	Bright vs. Dark	Bright vs. Neutral	Dark vs. Neutral
		MANOVA: Wilks’ Lambda: 0.878, F(20, 1472) = 9.804, *p* < 0.001, partial eta squared: 0.063								
Personality Traits										
	Self-esteem	4.12 (0.72)	3.58 (0.87)	3.98 (0.79)	3.97 (0.80)	32.91	<0.001	<0.001	0.006	<0.001
	Autonomy	3.24 (0.64)	2.77 (0.79)	3.08 (0.70)	3.09 (0.71)	31.16	<0.001	<0.001	<0.001	<0.001
Emotional and Mental Health										
	Psychological well-being	4.29 (0.59)	3.66 (0.87)	4.10 (0.70)	4.11 (0.71)	56.78	<0.001	<0.001	<0.001	<0.001
	Moods and emotional states	4.21 (0.77)	3.61 (0.95)	4.07 (0.84)	4.06 (0.85)	35.08	<0.001	<0.001	0.014	<0.001
Social Relationships										
	Relationships with parents and home	4.37 (0.76)	3.89 (0.91)	4.23 (0.81)	4.23 (0.82)	24.21	<0.001	<0.001	0.005	<0.001
	Peer relationships	4.31 (0.65)	3.81 (0.85)	4.08 (0.73)	4.12 (0.74)	35.65	<0.001	<0.001	<0.001	<0.001
	Social acceptance	4.53 (0.75)	4.24 (1.03)	4.53 (0.77)	4.49 (0.80)	10.76	<0.001	<0.001	<0.001	<0.001
Environmental and Contextual Factors										
	School environment	3.92 (0.71)	3.19 (0.93)	3.66 (0.83)	3.69 (0.84)	57.85	<0.001	<0.001	<0.001	<0.001
	Financial prospects	4.30 (0.87)	3.90 (1.11)	4.25 (0.93)	4.22 (0.94)	13.30	<0.001	<0.001	0.274	<0.001
Health and Physical Well-being										
	Physical well-being	4.13 (0.63)	3.53 (0.77)	3.95 (0.70)	3.96 (0.71)	53.06	<0.001	<0.001	<0.001	<0.001
Sleep (N)		486 M (SD)	186 M (SD)	774 M (SD)	1446 M (SD)					
		ANOVA: F(2, 1443) = 3.09, *p* < 0.001, partial eta squared: 0.052								
	Insomnia	6.17 (3.80)	9.45 (5.45)	6.32 (4.22)	6.67 (4.39)	45.65	<0.001	<0.001	<0.001	<0.001

**Table 2 jcm-13-06785-t002:** Pearson correlation between dark- and bright-side hypomania, insomnia, and health-related quality of life.

		Pearson Correlation	
Health-Related Quality of Life		Bright Side	Dark Side
Personality Traits			
	Self-esteem	0.059 *	−0.274 **
	Autonomy	0.150 **	−0.169 **
Emotional and Mental Health			
	Psychological well-being	0.161 **	−0.270 **
	Moods and emotional states	0.060 *	−0.307 **
Social Relationships			
	Relationships with parents and home	0.039	−0.247 **
	Peer relationships	0.221 **	−0.089 **
	Social acceptance	−0.010	−0.149 **
Environmental and Contextual Factors			
	School environment	0.137 **	−0.300 **
	Financial prospects	0.028	−0.155 **
Health and Physical Well-being			
	Physical well-being	0.154 **	−0.259 **
Sleep			
	Insomnia	−0.025	0.294 **

* The correlation is significant at the 0.01 level (2-sided). ** The correlation is significant at the 0.05 level (2-sided).

**Table 3 jcm-13-06785-t003:** Overview of the two multiple regression models with the bright and dark side of hypomania and health-related quality of life and insomnia.

		ANOVA					Model Summary				
		Sum of Squares	df	Mean of Sum	F	*p*	R	R^2^	R^2^ Corr	Standard Error of Estimator	Durbin–Watson Statistics
Dark-Side Hypomania	HRQOL	328.136	10	32.814	26.103	<0.001 **	0.389	0.151	0.146	1.121	1.989
	Insomnia	183.150	1	183.150	136.451	<0.001 **	0.294	0.086	0.086	1.158	1.950
Bright-Side Hypomania	HRQOL	171.274	10	17.127	11.396	<0.001 **	0.268	0.072	0.066	1.227	1.964
	Insomnia	1.473	1	1.473	0.915	0.339	0.025	0.001	0.000	1.269	1.935

** The correlation is significant at the 0.05 level (2-sided).

**Table 4 jcm-13-06785-t004:** Multiple regression with the bright side of hypomania and health-related quality of life scales.

		Non-Standardized Coefficients		Standardized Coefficient				
Bright-Side Hypomania		Coefficient Beta	Standard Error	Beta	T	*p*	Tolerance	VIF
HRQOL		0.647	0.268		2.417	0.016 *		
Personality traits								
	Self-esteem	−0.017	0.051	−0.011	−0.341	0.734	0.629	1.591
	Autonomy	0.067	0.058	0.037	1.151	0.250	0.606	1.650
Emotional and Mental Health								
	Moods and emotional states	−0.060	0.049	−0.040	−1.211	0.226	0.577	1.735
	Psychological well-being	0.141	0.089	0.079	1.580	0.114	0.252	3.973
Social Relationships								
	Relationships with parents and home	−0.140	0.052	−0.090	−2.695	0.007 *	0.563	1.777
	Peer relationships	0.326	0.053	0.190	6.164	<0.001 **	0.666	1.500
	Social acceptance	−0.058	0.044	−0.037	−1.336	0.182	0.832	1.201
Environmental and Contextual Factors								
	School environment	0.119	0.047	0.079	2.535	0.011 *	0.661	1.513
	Financial prospects	−0.063	0.041	−0.047	−1.555	0.120	0.699	1.431
Health and Physical Well-being								
	Physical well-being	0.098	0.086	0.055	1.145	0.252	0.276	3.629

* The correlation is significant at the 0.01 level (2-sided). ** The correlation is significant at the 0.05 level (2-sided).

**Table 5 jcm-13-06785-t005:** Multiple regression with the dark side of hypomania and health-related quality of life scales.

		Non-Standardized Coefficients		Standardized Coefficient				
Dark-Side Hypomania		Coefficient Beta	Standard Error	Beta	T	*p*	Tolerance	VIF
HRQOL		4.459	0.244		18.247	<0.001 **		
Personality traits								
	Self-esteem	−0.139	0.046	−0.091	−3.007	0.003 *	0.629	1.591
	Autonomy	−0.046	0.053	−0.027	−0.863	0.388	0.606	1.650
Emotional and Mental Health								
	Moods and emotional states	−0.208	0.045	−0.146	−4.593	<0.001 **	0.577	1.735
	Psychological well-being	−0.029	0.081	−0.017	−0.359	0.720	0.252	3.973
Social Relationships								
	Relationships with parents and home	−0.082	0.048	−0.055	−1.725	0.085	0.563	1.777
	Peer relationships	0.127	0.048	0.078	2.636	0.008 *	0.666	1.500
	Social acceptance	−0.096	0.040	−0.063	−2.404	0.016 *	0.832	1.201
Environmental and Contextual Factors								
	School environment	−0.258	0.043	−0.178	−6.005	<0.001 **	0.661	1.513
	Financial prospects	−0.012	0.037	−0.009	−323	0.747	0.699	1.431
Health and Physical Well-being								
	Physical well-being	0.028	0.079	0.016	−0.359	0.720	0.276	3.629

* The correlation is significant at the 0.01 level (2-sided). ** The correlation is significant at the 0.05 level (2-sided).

## Data Availability

Data might be made available under the following conditions: 1. Only an internationally recognized senior researcher can ask for the data set. 2. The scientific profile of the senior researcher is easily retrievable on the homepage of the institution. 3. The senior researcher contacts the corresponding author via her/his institutional email address (no @gmail.com or similar). 4. The senior researcher formulates clear-cut hypotheses; such hypotheses transparently describe the reasons as to why the data set should be provided. 5. The senior researcher describes credibly how the data set is securely stored on an institutional server, which is not accessible to a third party. 6. The senior researcher declares that the data set by no means is shared with a third party.

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
