# Peer review of "Illuminating Hypomania in Early Adolescence: Associations Between Dark-Side and Bright-Side Hypomania, Insomnia, and Health-Related Quality of Life"

_jcm, 2024, doi:10.3390/jcm13226785_

Round 1
Reviewer 1 Report
Comments and Suggestions for Authors
To the Authors:
Page 3: Cite original paper for this reference [13]
Page 8: Typo: (wwhich)
Page 9: Typo: ....no hypomania..
Page 11: Suggest authors to rephrase the sentence: The study protocol was carried out by the Declaration of Helsinki, and the local ethics committee approved the study (EKNZ code: 72/10 2010) to read
The study protocol was carried out in accordance to the Declaration of Helsinki, and the local ethics committee approved the study (EKNZ code: 72/10 2010)
Page 10: Participants
More information is needed regarding the participants. How were they selected? Was there any inclusion/exclusion criteria. How did the selection approach affect selection bias?
Also more information is needed on how the schools were selected. What informed the selection of the three districts? How were the five schools selected from the three districts? How are they representative of the target population? Was the school sampling based solely on convenience? What is the sociodemographic of these schools? Are these characteristics of these schools typical of the target population. This information might help contextualize the findings and external validity of the study.
Hypomania symptoms are time-changing variables and findings should be interpreted with caution, especially because of the nature of the study design and participant population (non-clinical adolescents).
To correctly contextualize and interpret the correlation between the school environment and the study outcomes, better description of the school districts and sociodemographic of the schools and district would be helpful.
Author Response
Please see the detailed point-by-point-response attached as a separate file. Thank you!

Reviewer 2 Report
Comments and Suggestions for Authors
Thank you for the opportunity to review this paper. This is a timely and important issue to explore. The aim of the study was to explore the relationships between bright and dark side hypomania, insomnia, and various dimensions of health-related quality of life (HRQOL), such as self-esteem, family and peer relationships, social acceptance, and autonomy.
Several comments and suggestions for the authors.
- The authors should standardize the reference list in the text (number with reference in square brackets, not name and year), the manuscript is messy in this respect.
- When was the research conducted (year and time frame not specified)?
- Please provide inclusion and exclusion criteria.
- Can the authors provide the Cronbach’s alpha value for all questionnaires?
- At the beginning of the discussion it is unnecessary to repeat the aim of the research.
- The text of the manuscript should be checked for punctuation, double spaces, etc.
- Research articles usually do not use the word „we”, „our” and regularly use passive verbs.
Author Response

(The authors gave the same response as above.)

Reviewer 3 Report
Comments and Suggestions for Authors
Eisenhut at al. present an interesting study here they tried to explore the relationships between bright and dark side hypomania, insomnia, and various dimensions of health-related quality of life (HRQOL) in 1,475 participants in mid-adolescence aged between 11 and 16 years- They could show that bright and dark side hypomania were associated with a broad, though specific variety of aspects of HRQOL and insomnia.
Minor issues:
1) In the introduction authors should add the information on the prevalence of bipolar disorders as this disease has the small prevalence of <1% in the global population (and in adolescents lower)
2) Authors use the classification of Hantouche for dark side and bright side hypomania. The classification sounds very interesting but is very old (2003). Can authors please explain how relevant this classification for the new studies is?
Author Response

(The authors gave the same response as above.)

Round 2
Reviewer 2 Report
Comments and Suggestions for Authors
Thank you for making the corrections, I have no further comments.
Author Response
Again, we thank Reviewer #2 for helping us to improve the quality of the second revision. Thank you!
